# Arbuscular Mycorrhizal Fungi Improve the Growth, Water Status, and Nutrient Uptake of *Cinnamomum migao* and the Soil Nutrient Stoichiometry under Drought Stress and Recovery

**DOI:** 10.3390/jof9030321

**Published:** 2023-03-05

**Authors:** Xuefeng Xiao, Xiaofeng Liao, Qiuxiao Yan, Yuangui Xie, Jingzhong Chen, Gelin Liang, Meng Chen, Shengyang Xiao, Yuan Chen, Jiming Liu

**Affiliations:** 1Forestry College, Guizhou University, Guiyang 550025, China; 2Guizhou Academy of Science, Guiyang 550001, China; 3College of Resources and Environmental Engineering, Guizhou University, Guiyang 550025, China; 4The Land Greening Remediation Engineering Research Center of Guizhou Province, Guiyang 550001, China

**Keywords:** AMF, *Cinnamomum migao*, drought stress, karst region, stoichiometry

## Abstract

Drought greatly influences the growth and ecological stoichiometry of plants in arid and semi-arid regions such as karst areas, where *Cinnamomum migao* (*C. migao*) is an endemic tree species that is used as a bioenergy resource. Arbuscular mycorrhizal fungi (AMF) play a key role in nutrient uptake in the soil-plant continuum, increasing plant tolerance to drought. However, few studies have examined the contribution of AMF in improving the growth of *C. migao* seedlings and the soil nutrient stoichiometry under drought-stress conditions. A pot experiment was conducted under natural light in a plastic greenhouse to investigate the effects of individual inoculation and Co-inoculation of AMF [*Funneliformis mosseae* (*F. mosseae*) and *Claroideoglomus etunicatum* (*C. etunicatum*)] on the growth, water status, and nutrient uptake of *C. migao* as well as the soil nutrient stoichiometry under well-watered (WW) and drought-stress (DS) conditions. The results showed that compared with non-AMF control (CK), AM symbiosis significantly stimulated plant growth and had higher dry mass. Mycorrhizal plants had better water status than corresponding CK plants. AMF colonization notably increased the total nitrogen and phosphorus content of *C. migao* seedlings compared with CK. Mycorrhizal plants had higher leaf and stem total carbon concentrations than CK. The results indicated that AM symbiosis protects *C. migao* seedlings against drought stress by improving growth, water status, and nutrient uptake. In general, the *C. migao* seedlings that formed with *C. etunicatum* showed the most beneficial effect on plant growth, water status, and nutrient uptake among all treatments. In the future, we should study more about the biological characteristics of each AMF in the field study to understand more ecological responses of AMF under drought stress, which can better provide meaningful guidance for afforestation projects in karst regions.

## 1. Introduction

Water is an essential ingredient for life, being necessary for plant growth and yield, especially in karst regions where plants are often exposed to periods of water shortage (drought stress) [1]. Drought is a recurring global climatic phenomenon and is among the most frequent natural disasters in many regions of the world [2]; it reduces soil nutrients [3] and uptake of plant nutrients [4,5], thereby altering the C:N:P stoichiometry of plants and decoupling biogeochemical cycles via the reduction of plant growth or water use and nutrient absorption [6,7,8].

Ecological stoichiometry has been widely studied in terrestrial ecosystems, as it can provide new ways to understand how plants and soil will respond to global climate change [9]. This analysis examined the correlations and relationships between soil-limiting elements and plant nutrients [10], indicating how the balance of energy and elements influences living organisms and their ecological interactions [11,12]. The dynamic responses of carbon (C), nitrogen (N), and phosphorus (P) stoichiometry of soil and plants to short- and long-term drought conditions is a current focus of ecological research that has been explored across many regions and ecosystems [12,13]. However, studies of C, N, and P ecological stoichiometry during drought stress in karst regions have rarely been reported.

Karst is a unique type of ecosystem that originates from limestone, dolomite, and carbonate rocks and is characterized by its underground drainage systems and rocky desertification [14]. Drought stress and nutrient deprivation frequently occur, severely limiting plant nutrient and water absorption and productivity [15,16] in karst regions, where *C. migao* is an endemic tree species that only exists in the dry-hot valleys of the transition zone of the Guizhou, Guangxi, and Yunnan Provinces of China [17]. The wood of *C. migao* is used as a bioenergy resource, and the tree is also used as an ethnic medicinal plant [18]. However, many natural populations of *C. migao* have disappeared because of allelopathy and autotoxicity [19,20]. Additionally, *C. migao* is very sensitive to drought stress [21,22]. Geological drought aggravates the survival crisis of *C. migao* in fragile karst areas, where water shortages and nutritional deficiencies are key limiting factors for plants [23].

AMF are beneficial microorganisms in the soil associated with the roots of 70–90% of terrestrial plant species [24,25], playing a significant role in ecosystems through water supplying and nutrient cycling [26,27], processes that strongly influence biogeochemical cycles of C, N, and P [28]. AMF acquire nutrients from the soil and then transfer them to the host plants in exchange for photosynthetically fixed carbon (C) [29,30]. Simultaneously, 5–10% of photosynthetically of host plants is allocated to the AMF partner [31]; when soil P is limited, roots and AMF selectively allocate more C and P to each other [32]. Under drought stress, AM fungi can absorb and transport water and N to host plants more efficiently through hypha [29]. AMF enhance their host plants’ tolerance to biotic and abiotic stresses, especially drought [33,34]. Studies have demonstrated that AMF can increase plant yield and help plants to resist the biotic and abiotic stresses that occur in agriculture and forestry [35,36], making the symbiotic interactions of plants with mycorrhizal fungi agriculturally and ecologically important [37,38,39]. Several studies have revealed that AMF are abundant in karst areas and that they can form a symbiosis with a range of *Lauraceae* species, including *C. migao* [40,41,42]. However, little is known about the effects of AMF on *C. migao* growth, water status, or C, N, and P ecological stoichiometry under drought stress and recovery in karst areas. The purposes of our study were to evaluate the effect of AMF on the growth, water status, and nutrient uptake of *C. migao*. We hypothesized that: (i) AMF inoculation can positively affect the growth and drought resistance of *C. migao* under drought-stress conditions. (ii) AMF inoculation can positively contribute to nutrient uptake, especially P of *C. migao*.

## 2. Materials and Methods

### 2.1. Study Area

We conducted the experiment from early April to early September 2021 in a plastic greenhouse under natural light with a day/night mean temperature of 24/16 °C, mean humidity of 57.3%, and average illumination: 1178.6 h·y^−1^. The plastic greenhouse was located at the Guizhou University in Guiyang City, Southwest China (26°340′ N, 106°420′ E; elevation: 242–1020 m). The area has a subtropical monsoon climate, with an annual mean temperature of 16.4 °C. The average annual rainfall is 1000–1400 mm, with 157 d of rainfall and 1000–1400 h of sunshine.

### 2.2. Growth Substrate, AMF, and Seeds

Field soil collected from karst areas with severe rocky desertification (25°64′ N, 107°07′ E; elevation: 790.28 m) was used as a growth substrate for the experiment. The growth substrate was sieved (2 mm) and then sterilized for 4 h at 121 °C and 0.14 MPa to eliminate all microorganisms. The growth substrate had the following chemical characteristics: pH 7.09 ± 0.56 (measured in water, 1:5 *w*/*v*); SOM (24.06 ± 1.40 g·kg^−1^); total nitrogen content (3.30 ± 0.13 g·kg^−1^); total phosphorus content (0.99 ± 0.05 g·kg^−1^); total potassium content (16.02 ± 1.80 g·kg^−1^). Then, 8 kg of sterilized soil was used to fill the pots (30 cm length × 20 cm height × 20 cm width).

Mycorrhizal fungal inocula of *F. mosseae* and *C. etunicatum* consisting of spores, soil, hyphae, and infected jowar root fragments were provided by the Institute of Plant Nutrition and Resources, Beijing Academy of Agriculture and Forestry Sciences, Beijing, China. The initial spore concentration of the inoculum was 120 spores/g. In total, 5 g of each inocula with white clover (*Trifolium repens*) seeds was placed in a pot (22.5 cm diameter × 22.5 cm height) filled with 2 kg sterilized sand (121 °C, 0.14 MPa for 4 h), respectively. Then, the pot was placed in a temperature incubator (25 °C, 75% relative humidity, 3000 lx (day/night, 16 h/8 h)) for 4 months. After removing the above-ground parts of the plants and the top 2 cm of topsoil, soil and underground parts of plants were collected as inocula. The inocula consisted of 70 spores/g, soil, hyphae, and infected white clover root fragments.

Seeds of *C. migao* were collected from healthy adult trees in a karst forest (25°26′40″ N, 106°31′51″ E; 667 maltitude) of Guizhou Province, China, in November 2020 and then were surface sterilized with 5% NaClO for 10 min, rinsed 5 times with distilled water, and sown in plastic seed trays with 200 g sterilized sand (121 °C, 0.14 MPa for 4 h) in a temperature-controlled incubator (day/night, 25 °C/20 °C, 80% relative humidity) for germination for 4 months. Seedlings were irrigated with sterile water once a day before transplanting into pots.

### 2.3. Experimental Design

The experimental treatments consisted of two factors: two soil water regimes [well-watered (WW) and drought stress (DS)] and four mycorrhizal treatments, individual and Co-inoculation of *F. mosseae* and *C. etunicatum*, and a sterilized inoculum (121 °C, 0.14 MPa for 4 h to prevent spontaneous mycorrhization) of *F. mosseae* and *C. etunicatum* was evenly applied into the substrate as the non-AMF control (CK). To prevent the unexpected death of the seedlings that would influence the experiment, we prepared 30 replicates (pots) per treatment. In total, there were 240 pots (plants) (one plant per pot). Either 20 g (dry wt) mycorrhizal with nearly 1400 spores (*F. mosseae* or *C. etunicatum* or Co-inoculation evenly) inoculum or sterilized inoculum was placed 5 cm under the soil surface such that it was in direct contact with the seedling roots.

This was a dynamic experiment conducted in three stages: all of the plants were well-watered for the first 90 days prior to stress (PS) to ensure that the AM colonization rate reached a stable point, then they were subjected to drought stress (SS) for the next 30 days (seedlings began wilting), and then rewatered (REC) for the next 30 days (seedlings resumed growth). During the WW treatment, deionized water was supplied to keep the relative soil water content at 75% of field capacity by weighing the pots every day. During the DS treatment, the pots were without any water supply until the seedlings began wilting.

### 2.4. Specimen Collection

At the end of each experiment stage, six pots were randomly selected to determine mycorrhization rate, plant growth, water status, gas-exchange parameters, plant nutrients, and soil nutrients. After removing the top 2 cm of topsoil, the firmly attached soil (1–2 mm distance from the root) was collected and regarded as rhizosphere soil, respectively. There were six replicates of each treatment at each experiment stage.

### 2.5. Plant Property Analysis

#### 2.5.1. AMF Colonization Rate

Roots from six plants were collected, washed gently under running tap water, and then rinsed with distilled water. Then, a subsample of 1 g was taken from the middle part of each root and cut into 1 cm-long segments that were bleached with 10% (*w*/*v*) KOH for 30 min at 90 °C and acidified in 1% HCl for 15 min, and then dyed with 0.05% (*w*/*v*) trypan blue in lactophenol [43]. Two hundred root segments per treatment were examined under an optical microscope (CX43, Olympus, Tokyo, Japan) to determine the mycorrhization rate according to the following formula [44]:AMF colonization (%) = (root length colonized/total root length observed) × 100.

#### 2.5.2. Plant Growth and Water Status Analysis

Plant height was measured with a ruler, and stem diameter was measured with a Vernier caliper. The fifth fully expanded leaf (from the apex) of each seedling was selected for measurement of the leaf area using a Handheld Leaf Area Meter (YMJ-B, Top, Hangzhou, China). After harvest, the leaves, stems, and roots were placed in an oven at 105 °C for 30 min and then dried at 80 °C for 7 days to constant mass, and the dry mass was determined using a precision analytical balance. The RWC of leaves was measured following [45]:RWC (%) = (FW − DW)/(SW − DW) × 100,
where FW is the fresh weight, DW is the dry weight, and SW is the turgid mass (mass after leaf samples were soaked in distilled water for 48 h).

The net photosynthetic rate (PN) and transpiration rate (E) of the fifth fully expanded leaf (from the apex) of each seedling were measured with a portable photosynthesis system Li-6800 (LiCor, Lincoln, NE, USA). The measurements were performed under approximately photosynthetically active radiation of 1500 μmol·m^−2^·s^−1^; the CO_2_ concentration in the sample chamber was 400 μmol·mol^−1^ from 08:30 to 11:00 h. Water-use efficiency (WUE) was calculated as the ratio PN/E. The water potential of the fifth fully expanded leaf (from the apex) of each seedling was measured with a System Analysis of Plant Stress (SAPS II, SEC, CA, USA).

#### 2.5.3. Nutrient Analysis

The dried samples mentioned above were manually crushed and sieved (0.2 mm) and then stored for nutrient analysis. The total concentration of carbon (SOM) in plants was measured by the potassium dichromate oxidation method [46]. The crushed and sieved (0.2 mm) dried samples were digested via the H_2_SO_4_-H_2_O_2_ method in a digesting block at 350 °C [47] before total N and total P measurements. The total concentration of nitrogen (TN) in the plants was measured by the indophenol blue colorimetry method and assayed at 625 nm; similarly, the total concentration of phosphorus (TP) in plants was measured by the vanadium molybdate yellow colorimetric method at 450 nm [48,49].

### 2.6. Soil Property Analysis

The soil samples were air dried and sieved (0.2 mm) under indoor temperature conditions to remove the roots for measuring soil properties. SOM of soil (soil SOM) was measured by the potassium dichromate oxidation method [50]. Soil total nitrogen content (soil TN) was measured by the indophenol blue colorimetry method, and digests produced according to the H_2_SO_4_-H_2_O_2_ method were assayed at 625 nm [48,49]. Soil total phosphorus content (soil TP) was measured by molybdenum blue colorimetry [51]. 

### 2.7. Statistical Analyses

All data were presented as the average of six replications of each treatment and were expressed as means ± standard error (SE). All data were subjected to a two-way analysis of variance (ANOVA) using a Tukey HSD post hoc test (*p* < 0.01; *p* < 0.05) using the SPSS 21.0 statistical program (SPSS Inc., IL, USA). The homogeneity of variance was verified before performing the ANOVA, and the data were logarithmically transformed when required. Origin 2021 (Origin Lab, Northampton, MA, USA) was used for drawing and processing. Then, principal component (PCA) analyses were performed to discriminate the performed to detect the growth, water status, and nutrient uptake of *C. migao* to drought stress inoculated with AMF during the whole experiment stages.

## 3. Results

### 3.1. AMF Colonization Rate

No colonization was observed in the CK treatments. During the experiment, AMF colonization was significantly affected by the type of AMF and water treatment (*p* < 0.05) (Figure 1). AMF colonization rate of *C. etunicatum* plants was significantly higher than that of *F. mosseae* and Co-inoculation plants regardless of the water regime. AMF colonization rate of *C. migao* seedlings was significantly increased under drought stress for each AMF except for *F. mosseae.*

### 3.2. Plant Water Status

RWC of *C. migao* leaves was significantly increased by AMF compared to CK (*p* < 0.05). RWC was significantly affected by drought stress only in the SS stage (Appendix A). In the SS stage, the RWC under WW conditions was significantly higher than that under DS conditions. Overall, during the experimental period, the water potential of *C. migao* leaves showed a trend of first decreasing and then increasing under DS conditions (Figure 2). However, there was hardly any change in the water potential for all treatments under WW conditions.

By the end of the PS stage, WUE was significantly affected by AMF, and the WUE of *C. etunicatum* plants was significantly higher than those of other treatments (Appendix A). By the end of the SS stage, AMF actively raised WUE under DS conditions compared to CK. However, the differences between treatments were significant under DS conditions.

### 3.3. Plant Growth

The plant height of *C. migao* seedlings was positively and significantly influenced by AMF (Figure 3; Table 1). Plant heights of all treatments under DS conditions were significantly lower than those under WW conditions. The stem diameter of AMF plants was significantly greater than that of CK during the experiment. Plants colonized by *C. etunicatum* exhibited significantly the greatest stem diameter among all treatments (Table 1). Drought had little effect on the stem diameter except for DS-Co-inoculation. In the PS stage, AMF had little effect on the leaf area. In the SS and REC stages, drought had little effect on leaf area, but AMF plants had significantly greater leaf area than CK, especially for *C. etunicatum*.

The dry mass of AMF plants was significantly higher than that of CK throughout the experimental period. In the PS stage, *C. migao* seedlings colonized by *C. etunicatum* exhibited significantly greater leaf dry mass (195.95%), stem dry mass (175.41%), and root dry mass (91.94%) compared with the CK seedlings. In the SS and REC stages, drought stress had little effect on the dry mass of *C. migao* seedlings (Appendix A). Seedlings inoculated with the *C. etunicatum* had significantly greater leaf dry mass (122.66%), stem dry mass (84.69%), and root dry mass (50.50%) compared with the CK. 

### 3.4. Plant C, N, and P Contents

The total concentration of carbon in leaves (leaf SOM) of AMF plants was significantly higher than that of CK (*p* < 0.05) (Appendix A). The leaf SOM under DS conditions first decreased and then increased with the progress of the experiment, while CK decreased throughout the experiment; otherwise, the leaf SOM under WW conditions showed a rising trend. Interestingly, the total concentration of carbon in stems (stem SOM) of CK plants was significantly higher than that of AMF plants (*p* < 0.05) (Appendix A). The stem SOM of all treatments showed a significant increasing trend, except for DS-CK and Co-inoculation. The total concentration of carbon in roots (root SOM) showed a trend of first decreasing and then increasing under DS conditions. However, there was little change in the root SOM of each treatment under WW conditions.

The total concentration of nitrogen in leaves (leaf TN) of Co-inoculation plants was significantly higher than in other treatments, regardless of water status (Appendix A). As the experiment proceeded, the leaf TN of all treatments under DS conditions showed a trend of first decreasing and then increasing, except for DS-CK. The total concentration of N in stems (stem TN) of CK plants was significantly greater than those of AMF plants. The stem TN of all treatments under DS conditions decreased as the experiment proceeded. During the experiment stages, the stem TN of CK was always greater than those of AMF plants, regardless of water status (Appendix A). The total concentration of N in roots (root TN) of *C. migao* seedlings colonized by *C. etunicatum* exhibited significantly greater root TN values of 30.14%, 43.50%, and 32.14% compared with the CK, respectively.

During the experiment, the total concentration of phosphorus (TP) of *C. migao* seedlings was positively and significantly influenced by AMF (Appendix A). Drought stress decreased the leaf and stem TPs of all treatments except for CK but increased the root TP as the experiment proceeded.

### 3.5. Plant C:N:P Ratios

The analysis of variance revealed little differences among the treatments in terms of leaf C:N ratio and drought stress had little effect on the leaf C:N ratio (Appendix A). The stem C:N ratio showed a gradually increasing trend with the progress of the experiment, except for Co-inoculation. Drought stress increased the stem C:N, and AMF significantly decreased the ratio. During the experiment stage, seedlings inoculated with *C. etunicatum* had lower stem C:N than other treatments. Drought stress and AMF significantly decreased the root C:N ratio. 

Overall, the changes in the magnitude of the leaf C:P ratio were small compared to those in the root C:P and stem C:P (Appendix A). Inoculation with AMF significantly decreased the C:P ratios of the plants under both DS and WW conditions compared to the CK treatment (*p* < 0.05). The leaf C:P and stem C:P both showed a gradually increasing trend with the progress of the experiment, while the root C:P showed a trend of first decreasing and then increasing under DS conditions; otherwise, there were no significant change trends under WW conditions.

Overall, the leaf N:P and stem N:P responses to moisture and AMF were opposite (Appendix A). The root N:P was significantly decreased by AMF but was unchanged under drought stress except for CK, which was significantly decreased by drought stress. In the PS stage, the leaf N:P was lowest in the seedlings inoculated with *C. etunicatum*.

### 3.6. Soil C, N, and P Contents

During the experiment, the soil nutrition was positively and significantly influenced by AMF, except for the soil TN in the PS stage (Appendix A). The total concentration of C in the soil (soil SOM) under DS conditions decreased initially in the SS stage and then increased in the REC stage; otherwise, under WW conditions, there was little change as the experiment proceeded. The soil TN under DS conditions increased initially in the SS stage and then decreased in the REC stage; otherwise, under WW conditions, soil TN increased except for WW-CK as the experiment proceeded. There was little change in soil total P concentration (soil TP) during the experiment, and the soil TP of the CK treatment was significantly higher than in the AMF treatments, irrespective of water condition. The soil TP of the CK treatment was significantly higher than in the AMF treatments, irrespective of water condition.

### 3.7. Soil C:N:P Ratios

The soil C:N:P ratios were significantly influenced by AMF during the experiment, except for the C:N ratio in the REC stage (Appendix A). Overall, the soil C:N under DS conditions decreased initially and then markedly increased; otherwise, under WW conditions, the ratio showed a gradually increasing trend with the progress of the experiment. In the PS stage, the soil C:N of Co-inoculation was visibly lower than in other treatments. In the SS stage, the soil C:N ratio was notably reduced by drought stress, being decreased the most in the Co-inoculation treatment. In the REC stage, the soil C:N of the CK treatment was markedly higher under the DS and WW conditions. Drought significantly reduced the soil C:P ratio in the SS stage. In the REC stage, the soil C:P ratio of DS-CK was greater than in the AMF and WW-CK treatments, respectively. The soil N:P ratio under DS conditions increased initially and then decreased; the ratio under WW conditions gradually increased except for WW-Co-inoculation. The soil N:P ratio of CK was lower than in other treatments under DS and WW.

### 3.8. Relationships among Plant and Soil Nutrient Stoichiometries

The correlation analysis (Figure 4, Figure 5 and Figure 6) showed strong associations between different nutrient stoichiometries of plants and soil in different experimental stages. In the PS stage, the leaf and root TN were both significantly positively (r = 0.50; r = 0.34) correlated with the soil TN (Figure 4). The root TP was also observed to have a significant negative (r = −0.34) correlation with soil TP. The stem SOM was significantly positively correlated with the soil SOM (r = 0.79); the leaf SOM was significantly negatively correlated with the soil SOM (r = −0.48). The C:N ratio in leaves was significantly positively correlated with the C:N ratio in soil (r = 0.42), and the C:P ratios in leaves, stems, and roots were correlated with the C:P ratio in the soil (r = −0.33, −0.37, −0.35, respectively) (Figure 4). The N:P ratios in stems and roots also had significant negative correlations with the N:P ratio in soil (r = −0.32, −0.41). In the SS stage, the leaf TN was negatively correlated (r = −0.58) with the soil TN (Figure 5), and the root TN had a significant positive (r = 0.49) correlation with soil TN. The stem SOM was significantly positively correlated with the soil SOM (r = 0.85); otherwise, there was no significant correlation between the content of TP in plants and soil TP. The C:N ratio in roots showed a positive correlation with C:N in soil (Figure 5). The C:P ratios in leaves, stems, and roots showed negative correlations with the C:P ratio in soil (r = −0.49, −0.64, and −0.33, respectively). A significant negative correlation was also observed between the stem and root N:P and soil N:P (r = −0.35 and −0.43, respectively). In the REC stage, there was no significant correlation between plant TN and soil TN or between plant TP and soil TP (Figure 6). Only leaf SOM was significantly negatively correlated with soil SOM (r = −0.37). Overall, there were no significant correlations between plant C:N and soil C:N, or leaf C:P and soil C:P or leaf N:P and soil N:P. The C:P ratios in leaves and roots were significantly positively correlated with the soil C:P ratio (r = 0.60, 0.49). The stem N:P and root N:P ratios were both negatively correlated with the soil N:P ratio (r = −0.34, −0.49).

### 3.9. Multivariate Statistical Analysis

Principal component analysis (PCA) was performed to detect the growth, water status, and nutrient uptake of *C. migao* to drought stress inoculated with AMF (Figure 7) during the whole experiment stages. More than 59.1% of the variance of the PCA was explained by the first two components (PC1 and PC2); Axis 1 of the PCA explained 44.9% of the variation, and PC2 showed a variation of 14.2%, respectively. Further, PC1 thoroughly separated the CK and AMF treatments. AMF treatments were associated with a higher plant height, dry mass, water status, and nutrient uptake, except for stem SOM and TP. Conversely, CK treatments were related to variables associated with nutrient stoichiometry ratios and stem SOM and TP. In PC1, the plant growth parameters, including plant height, stem diameter, dry mass (leaf, stem, and root dry mass), and TN content (leaf, stem, and root TN), were the key factors. By contrast, the TP content (leaf, stem and root TP), Stem C:N, and Leaf C:P were the key factors in PC2. Since PC1 contributed more to explaining the variation and was more correlated with the AMF treatment, it is more meaningful to focus on the plant growth, water status, and TN content of leaf, stem, and root to measure the effect of AMF on the growth and drought resistance of *C. migao.*

## 4. Discussion

Drought can harm both plants and their AMF partners, thereby affecting the colonization of the roots [52]. The mycorrhizal colonization rate of *C. etunicatum* plants was the highest in all treatments. One reason is that *Glomus* species are typical of semi-arid ecosystems [40], even in karst areas, and are able to adapt and grow under drought stress [53,54,55]. Second, the main ecological function of *C. etunicatum* is to promote primary plant production and P absorption, especially in the karst regions, where P is the main limiting element. Thirdly, *C. etunicatum* is probably a native partner of *C. migao* and has been symbiotic for a long history [56]. Additionally, *C. etunicatum* may have longer hypha than other fungi for adapting to drought [57]. For each AMF, the mycorrhizal colonization rate was significantly increased under drought stress. It may be that drought enhances soil O_2_ [58], increases soil heterogeneity [59], and restricts nutrient availability [60] and mobility, providing more favorable conditions for the growth of aerobic microorganisms [61]. Moreover, when exposed to stress, plant roots can detect environmental signals and release more secondary metabolome to attract microorganisms [62]. Co-inoculation inoculation inhibited root fungal colonization relatively more than individual inoculation in our study. One reason may be that co-inoculation enhances interspecific competition and inhibits the increase of spore number in rhizosphere soil [63,64]. Another reason may be that the production of reactive oxygen species in host plants and their antioxidant defense systems may influence the sequential colonization of these two endophytic fungi [65,66]. Some studies have suggested that drought stress significantly decreased AM colonization, negatively affecting the AMF development of host plants [67]. However, as a whole, most studies have shown that drought stress increases the mycorrhizal colonization rate [68]. For the observed differences, it might be that different plants may respond differently to drought stress and AMF symbiosis [61,62,63]. 

WUE, RWC, and water potential are important indicators reflecting plant water status and metabolism [69]. In our study, AMF colonization significantly improved the leaf WUE and RWC of *C. migao* seedlings under both WW and DS treatments. Stomatal closure is one of the most primitive plant responses to drought stress, and AMF can improve cellular metabolites and stomatal conductance [61,70], significantly reduce leaf water potential, and slow down cell division and elongation, resulting in the accumulation of metabolites. Previous studies have also indicated that mycorrhizal plants often have higher RWC, WUE, and water potential compared to nonmycorrhizal plants [71,72]. The results indicated that AMF increased the ability of roots to absorb soil moisture and systemically modified plant gene expression [73], thus improving their water uptake and maintaining opened stomata and then enhancing dry matter production under drought stress [74,75]. AMF expand the absorption region of the host plant roots by the extensibility of hyphae [76] and regulation of stomatal conductance by hormonal signals; this optimization of the osmotic adjustment enhances the absorption of water [71].

Plant growth-promoting rhizobacteria, including AMF, can ameliorate drought stress and improve plant growth and agronomic sustainability [58,77,78]. There are numerous studies indicating that AMF can promote plant growth, for example, in maize [79], soybean [80], *Ephedra foliata* Boiss [81], *citrus* [82], *Cyclobalanopsis glauca* [83], and sunflowers [53], which indicates that mycorrhizal colonization relieved growth inhibition and improved water status under drought [84]. AMF can form extensive hyphal networks in host roots and improve the growth of host plants by promoting nutrient and water uptake [85]. Additionally, AMF can help to increase chlorophyll synthesis [86], production of ROS [87], mineral solubilization [88], and photosynthesis [89,90] of host plants, thereby reducing abiotic stress. Moreover, AMF can effectively increase the levels of Osmoprotectants in plant leaves and provide potential protection against abiotic stress [80] and recruit other microorganisms in the soil, thereby altering the rhizosphere microbial environment of the host plants and increasing mineralization [91].

Our findings showed that AMF significantly increased the total nitrogen and phosphorus contents in host plants. Nitrogen is a critical component for all enzymes, and phosphorus is essential for rRNA synthesis [92]. Thus, changes in phosphorus and nitrogen concentrations can affect the allocation of nutrients, life history strategies, and physiological functions of cellular components in plants faced with abiotic stress [93]. Previous research has shown that AMF symbionts can provide nearly 42% of the nitrogen to their host plants [94]. AMF have a higher affinity for NH_4_^+^ compared to plant roots [95] and thus can take up both forms of N, either NH_3_ or NH_4_^+^, via hyphae through exploring large volumes of soil, by entering soil pores or by transferring N from organic patches with the help of hydrolytic enzymes such as phosphatases, cellulases, and chitinases [32,96,97]. NH_3_/NH_4_^+^ is the preferential form of N released by the fungus and absorbed by the plant [98].

Many studies have concluded that AMF can significantly enhance phosphorus uptake and mineralization by host plants, especially in soils with low phosphorus levels [99,100]. Phosphorus content in soil is generally low in tropical and subtropical forests [101], especially in the karst area [102]. Under P-limiting conditions, plants tend to maximize the efficiency of phosphorus uptake by increasing the secretion of root carboxylates such as citrate and malate to increase available P [103]. Microbes such as AMF always make an effort to synthesize enzymes when the substrate is abundant and products are deficient relative to the demand [104] in the following ways: first, mycorrhizal hyphae lengths range from 2 to 35 mg^−1^ soil (20–1400 mm^−1^ root) and are controlled by nutrient levels, primarily by soil-P levels [105,106]. Second, AMF-symbiotic increases in P-uptake may be due to AMF releasing some microorganisms that chelate cations, such as phosphates and organic acids that combine with phosphates, ultimately improving the availability of phosphorus in the soil [107].

Our findings showed that AMF significantly improved the SOM in the leaves and roots of host plants but decreased the content in stems. SOM is an essential constituent for the synthesis of structural compounds of organs, such as lignin protecting roots and leaves [108]. In turn, roots and leaves have sink–source relationships, and the increase in SOM content of leaves promotes the transfer of photosynthetic products to roots for adapting to arid environments and improving defense capability [92].

The C:N:P ratio represents the efficiency of plant photosynthesis, indicating the assimilation ability of carbon by plants in the process of nutrient harvesting and reflecting the nutrient usage efficiency [109,110]. In our study, the results were consistent with recent studies showing that higher growth rates are related not only to lower C:P and C:N ratios but also to a lower N:P ratio [111]. Otherwise, higher C:P and C:N ratios decrease the fresh weight value of plants [112]. In addition, the N:P ratio is a key indicator for judging whether plant productivity is limited by environmental nutrients [113]. AMF inoculation is a key determinant of plant stoichiometry, as the interaction from the environmentally regulated level helps to improve the nutritional status of plants to resist damage from abiotic stress.

Soil C:N:P is an important parameter for measuring soil nutrient balance, and it is also an important indicator of the composition and quality of soil organic matter [114]. The ratio is also affected by the rate of decomposition driven by soil microorganisms [115]. The importance of AMF in the regulation of soil nutrient cycling has been increasingly acknowledged [116,117,118]. AMF can contribute to soil nutrient retention, especially under drought conditions, by minimizing losses that typically occur through leaching and gas emissions via several mechanisms. First, AMF can enlarge the nutrient interception zone by the extensibility of hyphae, thereby enhancing nutrient uptake of plants as well as mycorrhizal nutrient immobilization [114,119]. Second, AMF can reduce the volume of leachate by increasing the water uptake rate, thereby improving soil water-holding capacity and ultimately avoiding the loss of soil nutrients [35]. Last but not least, AMF can recruit other microorganisms, thereby altering the community structure of soil involved in the cycling of N [91,117] and P [120]. Soil C:N is generally inversely proportional to the rate of decomposition of organic matter [121]. C:P always reflects the availability of soil phosphorus, and the N:P ratio can be used as an effective indicator of nutrient limitation in ecosystems [115]. The average values of C:N, C:P, and N:P in Chinese soils are 11.9, 61, and 5.2, respectively [122]. In our study, the soil C:N ratio was slightly higher than the average value, but the C:P and N:P ratios were significantly lower, meaning that the decomposition rate of organic matter in karst regions is slightly lower than the average in China [123].

AMF always play a key role in nutrient uptake in the soil-plant continuum, especially in stressful environments with very low nutrient content, a process that reinforces the plant–soil feedback between plant belowground and aboveground ecosystem compartments [124]. In our study, the soil C:N ratios showed significant positive correlations with plant C:N, C:P, and N:P ratios. The soil N:P ratios showed negative correlations with the plant C:N, C:P, and N:P ratios. The soil C:P ratios showed similar relationships to soil C:P ratios except for C:N in the roots in the PS stage. The results were consistent with those of previous studies [125,126], indicating that there are strongly coupled relationships between plant and soil ecological stoichiometries. Stoichiometries have been used to study the mechanisms of plant adaption to environmental change [127]. The presence of microorganisms, especially AMF, has an irreplaceable role in the shifting and maintenance of plant and soil stoichiometry [128]. Previous studies indicated that AMF could stabilize community productivity under changing soil N:P by increasing the stoichiometric homeostasis of the plant community [129]. AMF associated with plants effectively provide plant nutrients, especially in low-fertility and stressed environments [130], consistent with our results. Therefore, maintaining the nutrient balance in *C. migao* by AMF may be an adaptive strategy in karst regions, as these are arid and barren environments.

## 5. Conclusions

In summary, the present study demonstrated that AMF can improve the growth, water status, and nutrient uptake, especially the TN content of the leaf, stem, and root of *C. migao* seedlings under drought stress and recovery. It is clear that AMF promotes phosphorus uptake for host plants so that more C and N can be allocated to the leaves and roots, and AMF always play a key role in the nutrient uptake of host plants by regulating soil nutrient contents under drought stress and recovery. It is obvious that *C. etunicatum* had the most beneficial effect on plant growth, water status, and nutrient uptake among all treatments. Many of the current studies have found this phenomenon, and few related studies were conducted with the underlying mechanisms still unknown. In the future, we should study more about the biological characteristics of each AMF and understand the ecological responses and effects of AMF under drought stress. We should also progress with more field studies, which can better provide detailed and meaningful guidance for afforestation projects in karst regions.

## Figures and Tables

**Figure 1 jof-09-00321-f001:**
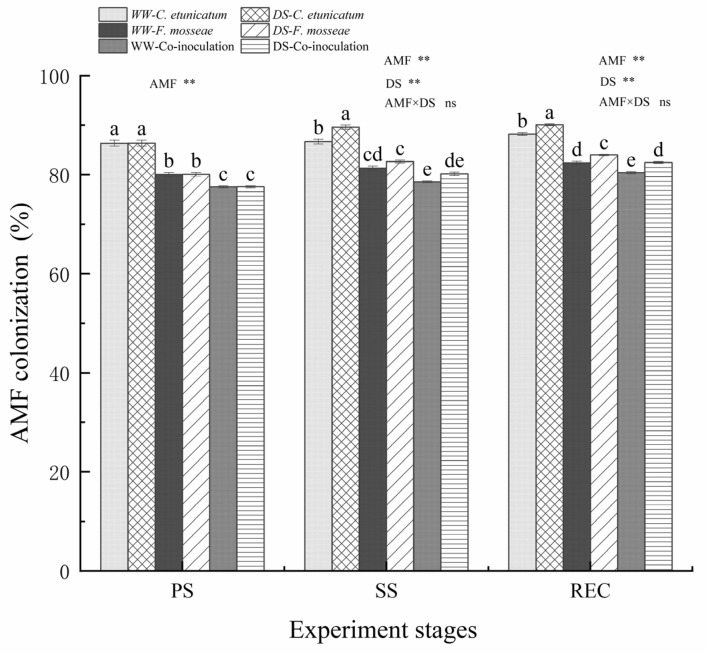
Effect of arbuscular mycorrhizal fungus (AMF) on the AMF colonization rate of *C. migao* seedlings measured at different experiment stages. Different letters (a, b, c, d, e) indicate a significant difference in Tukey’s post hoc test (** *p* < 0.01; ns, not significant) between all treatments. PS: prior stress, SS: subjected to drought stress; REC: rewatered; DS: drought stress; WW: well−watered. *C. etunicatum*, plants inoculated with *C. etunicatum*; *F. mosseae*, plants inoculated with *F. mosseae*; Co−inoculation, plants inoculated with *F. mosseae* and *C. etunicatum* Values are expressed as the mean ± SE (*n* = 6, which are treatment replicates).

**Figure 2 jof-09-00321-f002:**
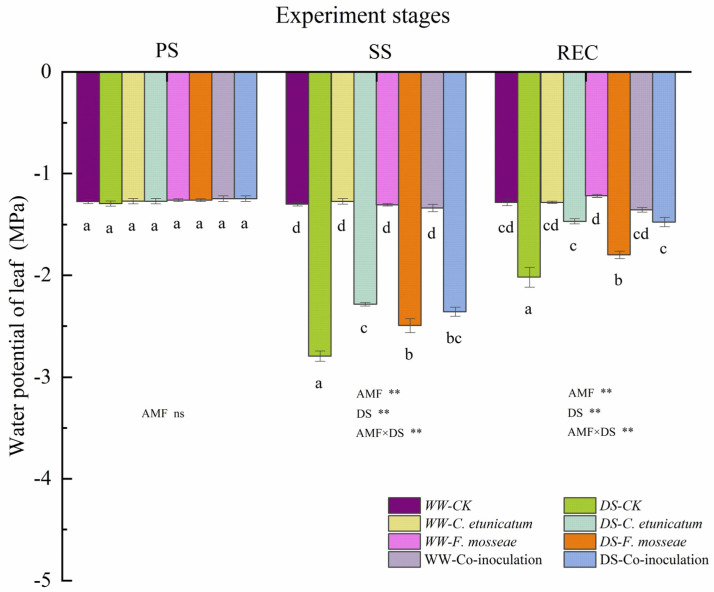
Effect of arbuscular mycorrhizal fungus (AMF)on the water potential of *C. migao* seedlings measured at different experiment stages. Different letters (a, b, c, d) indicate a significant difference in Tukey’s post hoc test (** *p* < 0.01; ns, not significant) between all treatments. PS: prior stress, SS: subjected to drought stress; REC: rewatered; DS: drought stress; WW: well−watered. CK, non−AMF control; *C. etunicatum*, plants inoculated with *C. etunicatum*; *F. mosseae,* plants inoculated with *F. mosseae;* Co−inoculation, plants inoculated with *F. mosseae* and *C. etunicatum* Values are expressed as the mean ± SE (*n* = 6, which are treatment replicates).

**Figure 3 jof-09-00321-f003:**
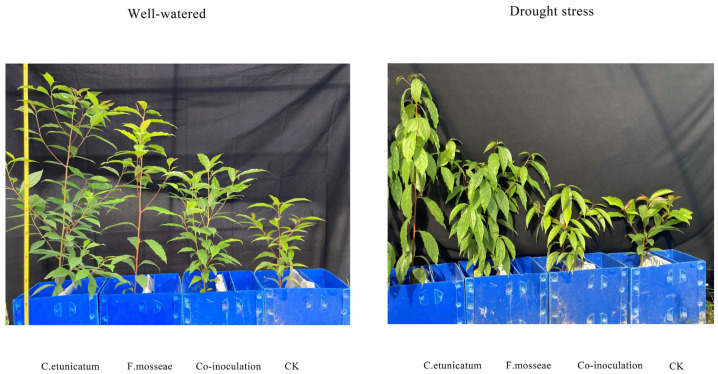
Effect of arbuscular mycorrhizal fungus (AMF) on the Plant height of *C. migao* seedlings. CK, non-AMF control; *C. etunicatum*, plants inoculated with *C. etunicatum*; *F. mosseae,* plants inoculated with *F. mosseae;* Co-inoculation, plants inoculated with *F. mosseae* and *C. etunicatum*.

**Figure 4 jof-09-00321-f004:**
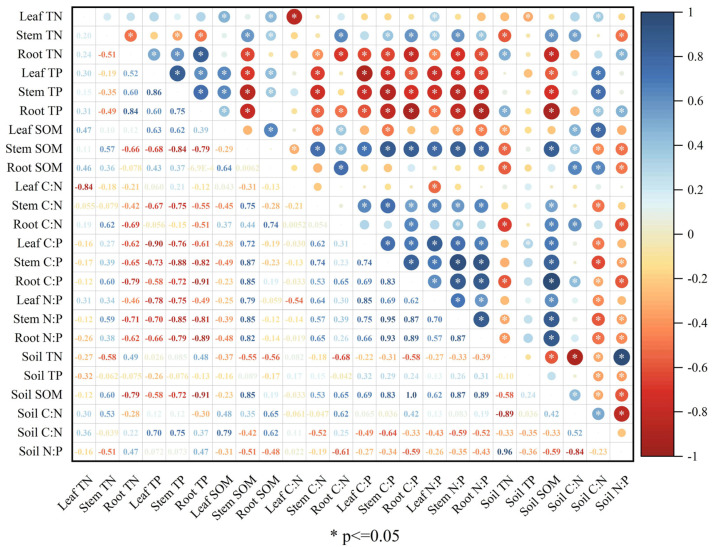
Spearman analysis of nutrition and stoichiometry of *C. migao* seedlings and soil in the PS stage. Leaf TN: the total concentration of nitrogen in leaves; Stem TN: the total concentration of nitrogen in stems; Root TN: the total concentration of nitrogen in roots; Leaf TP: the total concentration of phosphorus in leaves; Stem TP, the total concentration of phosphorus in stems; Root TP: the total concentration of phosphorus in roots; Leaf SOM: the total concentration of carbon in leaves; Stem SOM: the total concentration of carbon in stems; Root SOM: the total concentration of carbon in roots; C:N in leaf: carbon: nitrogen in leaves; C:N in stem: carbon: nitrogen in stems; C:N in root: carbon: nitrogen in roots; C:P in leaf: carbon: phosphorus in leaves; C:P in stem: carbon: phosphorus in stems; C:P in root: carbon: phosphorus in roots; N:P in leaf: nitrogen: phosphorus in leaves; N:P in stem: nitrogen: phosphorus in stems; N:P in root: nitrogen: phosphorus in roots; Soil TN: the total concentration of nitrogen in soil; Soil TP, the total concentration of phosphorus in soil; Soil SOM: the total concentration of carbon in soil; C:N in soil: carbon: nitrogen in soil; C:P in soil: carbon: phosphorus in soil; N:P in soil: nitrogen: phosphorus in soil. * *p* ≤ 0.05: significant correlation.

**Figure 5 jof-09-00321-f005:**
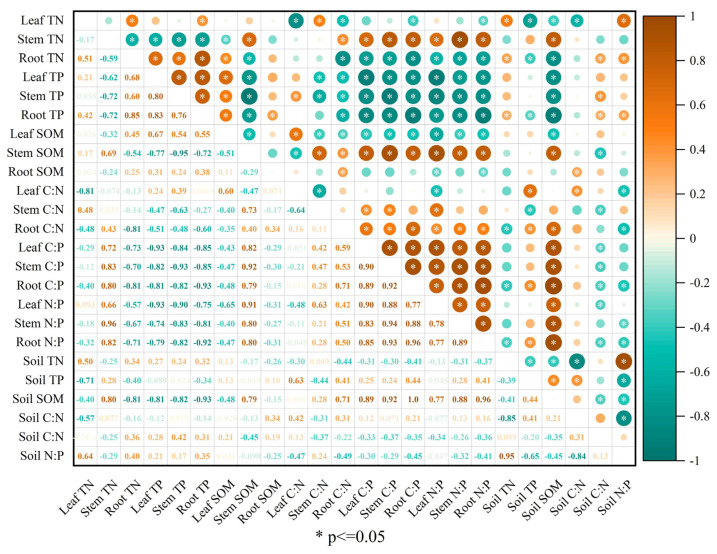
Spearman analysis of nutrition and stoichiometry of *C. migao* seedlings and soil in the SS stage. Leaf TN: the total concentration of nitrogen in leaves; Stem TN: the total concentration of nitrogen in stems; Root TN: the total concentration of nitrogen in roots; Leaf TP: the total concentration of phosphorus in leaves; Stem TP, the total concentration of phosphorus in stems; Root TP: the total concentration of phosphorus in roots; Leaf SOM: the total concentration of carbon in leaves; Stem SOM: the total concentration of carbon in stems; Root SOM: the total concentration of carbon in roots; C:N in leaf: carbon: nitrogen in leaves; C:N in stem: carbon: nitrogen in stems; C:N in root: carbon: nitrogen in roots; C:P in leaf: carbon: phosphorus in leaves; C:P in stem: carbon: phosphorus in stems; C:P in root: carbon: phosphorus in roots; N:P in leaf: nitrogen: phosphorus in leaves; N:P in stem: nitrogen: phosphorus in stems; N:P in root: nitrogen: phosphorus in roots; Soil TN: the total concentration of nitrogen in soil; Soil TP, the total concentration of phosphorus in soil; Soil SOM: the total concentration of carbon in soil; C:N in soil: carbon: nitrogen in soil; C:P in soil: carbon: phosphorus in soil; N:P in soil: nitrogen: phosphorus in soil. * *p* ≤ 0.05: significant correlation.

**Figure 6 jof-09-00321-f006:**
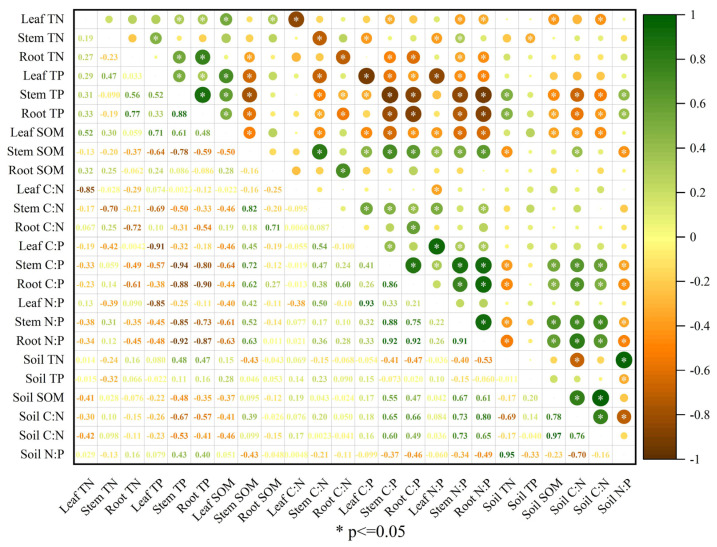
Spearman analysis of nutrition and stoichiometry of *C. migao* seedlings and soil in the REC stage. Leaf TN: the total concentration of nitrogen in leaves; Stem TN: the total concentration of nitrogen in stems; Root TN: the total concentration of nitrogen in roots; Leaf TP: the total concentration of phosphorus in leaves; Stem TP, the total concentration of phosphorus in stems; Root TP: the total concentration of phosphorus in roots; Leaf SOM: the total concentration of carbon in leaves; Stem SOM: the total concentration of carbon in stems; Root SOM: the total concentration of carbon in roots; C:N in leaf: carbon: nitrogen in leaves; C:N in stem: carbon: nitrogen in stems; C:N in root: carbon: nitrogen in roots; C:P in leaf: carbon: phosphorus in leaves; C:P in stem: carbon: phosphorus in stems; C:P in root: carbon: phosphorus in roots; N:P in leaf: nitrogen: phosphorus in leaves; N:P in stem: nitrogen: phosphorus in stems; N:P in root: nitrogen: phosphorus in roots; Soil TN: the total concentration of nitrogen in soil; Soil TP, the total concentration of phosphorus in soil; Soil SOM: the total concentration of carbon in soil; C:N in soil: carbon: nitrogen in soil; C:P in soil: carbon: phosphorus in soil; N:P in soil: nitrogen: phosphorus in soil. * *p* ≤ 0.05: significant correlation.

**Figure 7 jof-09-00321-f007:**
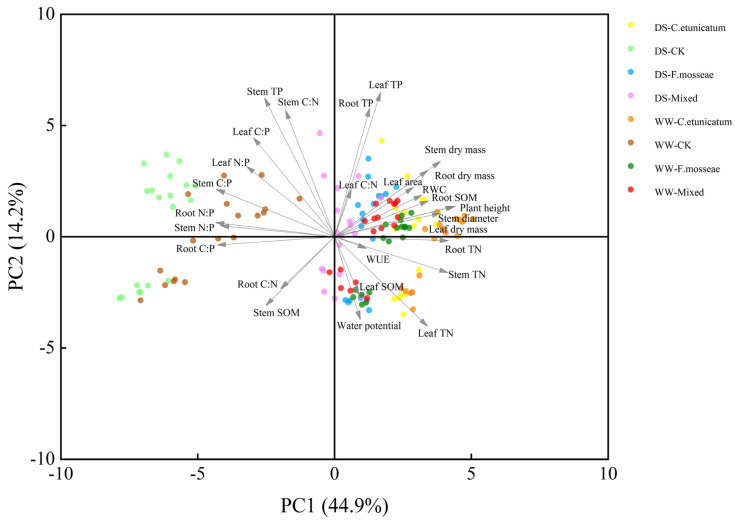
Principal component analysis (PCA) of *C. migao* inoculated with AMF under drought stress. DS: drought stress; WW: well−watered. *C. etunicatum*, plants inoculated with *C. etunicatum*; *F. mosseae,* plants inoculated with *F. mosseae;* Co−inoculation, plants inoculated with *F. mosseae* and *C. etunicatum*.

**Table 1 jof-09-00321-t001:** Effect of arbuscular mycorrhizal fungus (AMF) on the Plant height, Stem diameter, and Leaf area of *C. migao* seedlings measured at different experiment stages. Different letters (a, b, c, d, e, f) indicate a significant difference in Tukey’s post hoc test (** *p* < 0.01; * *p* < 0.05; ns, not significant) between all treatments. PS: prior stress, SS: subjected to drought stress; REC: rewatered; DS: drought stress; WW: well−watered. CK, non−AMF control; *C. etunicatum*, plants inoculated with *C. etunicatum*; *F. mosseae,* plants inoculated with *F. mosseae;* Co−inoculation, plants inoculated with *F. mosseae* and *C. etunicatum*. Values are expressed as the mean ± SE (*n* = 6, which are treatment replicates).

Water Regimes	AMF Status	Plant Height (cm)			Stem Diameter (mm)			Leaf Area (cm^2^)		
		PS	SS	REC	PS	SS	REC	PS	SS	REC
DS	CK	47.94 ± 2.26 ^c^	51.32 ± 2.62 ^d^	59.58 ± 1.82 ^f^	6.76 ± 0.19 ^d^	7.51 ± 0.30 ^e^	7.86 ± 0.22 ^d^	17.50 ± 1.33 ^a^	18.05 ± 1.25 ^b^	19.93 ± 1.53 ^c^
	*C. etunicatum*	93.04 ± 3.21 ^a^	98.86 ± 3.10 ^ab^	103.54 ± 2.33 ^b^	10.88 ± 0.33 ^a^	11.89 ± 0.10 ^ab^	12.17 ± 0.36 ^a^	22.37 ± 2.57 ^a^	23.99 ± 2.68 ^ab^	26.47 ± 2.43 ^ab^
	*F. mosseae*	82.18 ± 4.83 ^ab^	88.50 ± 4.73 ^c^	95.26 ± 3.17 ^c^	10.01 ± 0.34 ^ab^	10.92 ± 0.33 ^ab^	11.59 ± 0.37 ^ab^	19.10 ± 1.30 ^a^	20.28 ± 1.63 ^ab^	22.53 ± 1.53 ^bc^
	Co-inoculation	74.66 ± 2.86 ^b^	82.42 ± 3.65 ^c^	87.46 ± 2.27 ^d^	8.69 ± 0.62 ^bc^	9.25 ± 0.60 ^cd^	10.45 ± 0.47 ^c^	21.74 ± 2.67 ^a^	22.94 ± 2.68 ^ab^	24.95 ± 2.10 ^abc^
WW	CK	45.22 ± 2.09 ^c^	55.42 ± 1.33 ^d^	69.92 ± 2.60 ^e^	7.16 ± 0.14 ^cd^	7.84 ± 0.12 ^de^	8.53 ± 0.21 ^d^	17.99 ± 1.15 ^a^	18.60 ± 1.06 ^b^	19.84 ± 1.17 ^c^
	*C. etunicatum*	93.34 ± 3.15 ^a^	103.82 ± 3.32 ^a^	112.50 ± 1.23 ^a^	11.02 ± 0.34 ^a^	12.12 ± 0.05 ^a^	12.26 ± 0.14 ^a^	22.50 ± 2.54 ^a^	26.21 ± 1.82 ^a^	30.10 ± 1.07 ^a^
	*F. mosseae*	84.98 ± 4.69 ^ab^	92.46 ± 4.53 ^bc^	103.62 ± 2.22 ^b^	10.29 ± 0.37 ^ab^	11.41 ± 0.17 ^ab^	12.10 ± 0.23 ^ab^	19.79 ± 0.74 ^a^	21.93 ± 0.93 ^ab^	24.93 ± 0.60 ^abc^
	Co-inoculation	77.08 ± 2.46 ^b^	87.48 ± 2.63 ^c^	100.70 ± 1.41 ^bc^	8.85 ± 0.60 ^bc^	10.49 ± 0.43 ^bc^	11.22 ± 0.38 ^bc^	22.30 ± 2.64 ^a^	25.44 ± 2.36 ^a^	28.52 ± 2.27 ^a^
Significance										
AMF		**	**	**	**	**	**	ns	*	**
DS		ns	ns	**	ns	*	*	ns	ns	ns
AMF×DS		ns	ns	ns	ns	ns	ns	ns	ns	ns

## Data Availability

Not applicable.

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
