# Peer review of "Arbuscular Mycorrhizal Fungi Improve the Growth, Water Status, and Nutrient Uptake of Cinnamomum migao and the Soil Nutrient Stoichiometry under Drought Stress and Recovery"

_jof, 2023, doi:10.3390/jof9030321_

Round 1

Reviewer 1 Report (Previous Reviewer 1)

This interesting article has been carefully read and reviewed, there are still grammatical and typing errors, along with some issues in the content.

1.       In line 13, there was an unnecessary comma.

2.       Shorter forms of scientific names do not have to be shown, e.g. F. mosseae still can be known as Funneliformis mosseae without replacement.

3.       In line 22, missing “the” before “highest”.

4.       “CK” in line 23 and “TN” in line 24 should be shown in full, because this was the first time it appears.

5.       The Funneliformis mosseae and Claroideoglomus etunicatum should be emphasized as arbuscular mycorrhizal fungi in line 17.

6.       The conclusion in the abstract was inappropriate, because the roles of the AMF application is the target of this study, not the relationship between soil and plant.

7.       In line 52, the size of letter was bigger.

8.       The temperature in the greenhouse (24-16) is warmer than the average temperature in the area (16.4), does this mean that the greenhouse did not perform the actual condition?

9.       The temperature in the greenhouse is cool, while the C. migao was mentioned to be endemic in dry-hot valley, which altogether may affect the growth of the tree.

10.   Why there was too much autoclaving for soil samples, and how many days was the soil sterilized?

11.    In line 98, “white clover” is repeated.

12.   All of the plants, including the drought stress treatments, need to be watered well 90 days before stress, which did not express the actual drought stress.

13.   There were too many abbreviations in the article, REC, PN, R, RWC, etc. And they were continuously repeated along with their full forms throughout the article.

14.   In line 148 and 195, the size of letter was smaller.

15.   The PCA was not mentioned in the method section.

16.   The format of headings was not uniform throughout the article.

17.   In my opinion, pictures of C. migao treated with or without AMF under drought stress should be shown.

Author Response

Dear professor:

Re: Manuscript ID: jof-2187413 and Title: Arbuscular Mycorrhizal Fungi alter the growth, water status, and nutrient uptake of Cinnamomum migao and the soil nutrient stoichiometry under drought stress and recovery.

Hope this email finds you well.

Many thanks for your insightful comments and suggestions of the referees. Those comments are valuable and very helpful. We have read through comments carefully and have made corrections. Appended to this letter is our point-by-point response to your comments. The comments are reproduced and our responses are given directly afterward in a different color (blue).

  1. In line 13, there was an unnecessary comma.

Response: Thank you for the abstract suggestion. The unnecessary comma in line 13 was removed now.

  1. Shorter forms of scientific names do not have to be shown, e.g. F. mosseae still can be known as Funneliformis mosseae without replacement.

Response: Thanks for your insightful suggestions. We have replaced all long scientific names with shorter forms in the manuscript.

  1. In line 22, missing “the” before “highest”.

Response: Thanks very much for your insightful suggestions. We have added “the” before “highest” according to your suggestion.

  1. “CK” in line 23 and “TN” in line 24 should be shown in full, because this was the first time it appears.

Response: Thank you very much four your insightful comments. We have changed the “CK” to “non-AMF control (CK)” in abstract. And “TN” was replaced with “the total nitrogen content” too.

  1. The Funneliformis mosseae and Claroideoglomus etunicatum should be emphasized as arbuscular mycorrhizal fungi in line 17.

Response: Thank you for your suggestion very much. We have rewritten the sentence in lines 18-20  as follows:

“A pot experiment was conducted under natural light in a plastic greenhouse to investigate the effects of individual inoculation and co-inoculation of AMF [Funneliformis mosseae (F. mosseae) and Claroideoglomus etunicatum (C. etunicatum)]”.

  1. The conclusion in the abstract was inappropriate, because the roles of the AMF application is the target of this study, not the relationship between soil and plant.

Response: Thank you for your profound insight and precious comments. The abstract has been rewritten in lines 13-33 as follows:  

Abstract: Drought greatly influences the growth and ecological stoichiometry of plants in arid and semi-arid regions such as karst areas, where Cinnamomum migao (C. migao) is an endemic tree species that is used as a bioenergy resource. Arbuscular mycorrhizal fungi (AMF) play a key role in the nutrient uptake in the soil-plant continuum, increasing plant tolerance to drought. However, few studies have examined the contribution of AMF in improving the growth of C. migao seedlings and the soil nutrient stoichiometry under drought stress conditions. A pot experiment was conducted under natural light in a plastic greenhouse to investigate the effects of individual inoculation and co-inoculation of AMF [Funneliformis mosseae (F. mosseae) and Claroideoglomus etunicatum (C. etunicatum)] on the growth, water status, and nutrient uptake of C. migao as well as the soil nutrient stoichiometry under well-watered (WW) and drought stress (DS) conditions. The results showed that compared with non-AMF control (CK), AM symbiosis significantly stimulated plant growth and had higher dry mass. Mycorrhizal plants had better water status than corresponding CK plants. AMF colonization notably increased the total nitrogen and phosphorus content of C. migao seedlings compared with CK. Mycorrhizal plants had higher leaf and stem total carbon concentration than CK. The results indicated that AM symbiosis protect C. migao seedlings against drought stress through improving the growth, water status, and nutrient up-take. In general,the C. migao seedlings that formed with C. etunicatum showed most beneficial effect on the plant growth, water status, and nutrient uptake among all treatments. In the future, we should study more about the biology characteristics of each AMF in the field study to understand more ecological responses of AMF under drought stress, which can better provide meaningful guidance of afforestation projects in karst regions. 

  1. In line 52, the size of letter was bigger.

Response: Thank you very much for your precious suggestion. We are so sorry of our negligence. The size of the letters has been adjusted.

  1. The temperature in the greenhouse (24-16) is warmer than the average temperature in the area (16.4), does this mean that the greenhouse did not perform the actual condition?

Response: Thank you very much for your precious suggestion. We are so sorry some relevant description was missed because of our negligence. Actually, we conducted the experiment in a plastic greenhouse under natural light, which cannot control the temperature and humidity.

  1. The temperature in the greenhouse is cool, while the C. migao was mentioned to be endemic in dry-hot valley, which altogether may affect the growth of the tree.

Response: Thank you for the suggestion. In our future experiments, we will consider how the temperature affect the growth of C. migao you suggested. The purpose of our present study is that exploring which AMF can more effectively associate with C. migao seedlings and improve their drought resistance. And then, we will conduct a field study to verify the AMF we have selected in our present study so as to have more practical significance.

  1. Why there was too much autoclaving for soil samples, and how many days was the soil sterilized?

Response: Thank you very much for your precious suggestion. We are so sorry some inaccurate descriptions confused you because of our negligence. In lines 95-98, the sentence was reformatted as follows:

“Field soil collected from karst areas with severe rocky desertification (25◦64 N,107◦07 E; elevation: 790.28 m) was used as growth substrate for the experiment. The growth substrate was sieved (2 mm) and then sterilized for 4 h at 121°C and 0.14 MPa to eliminate all microorganisms.”

  1. In line 98, “white clover” is repeated.

Response: Thank you for your suggestion. We have divided removed the repeated “white clover” in line 107.

  1. All of the plants, including the drought stress treatments, need to be watered well 90 days before stress, which did not express the actual drought stress.

Response: Thank you for your profound insight and precious comments. In our future experiments, we will consider a constant drought stress treatment you suggested. Our present study is a dynamic one, all treatments are well watered 90 days prior to ensure that the AM fully inoculate with C. migao seedlings.

  1. In line 148 and 195, the size of letter was smaller.

Response: Thank you very much for your precious suggestion. We are so sorry of our negligence. The size of the letters has been adjusted.

  1. The PCA was not mentioned in the method section.

Response: Thank you very much for your precious suggestion. We are so sorry of our negligence. In lines 197-199, PCA was added in the method section as follows: “Then, principal component (PCA) analyses were performed to discriminate the performed to detect the growth, water status, and nutrient uptake of C. migao to drought stress inoculated with AMF during the whole experiment stages.”

  1. The format of headings was not uniform throughout the article.

Response: Thank you very much for your precious suggestion. We are so sorry of our negligence. The format of headings has been adjusted now.

  1. In my opinion, pictures of C. migao treated with or without AMF under drought stress should be shown.

Response: Thank you for your profound insight and precious comments. The pictures of C. migao treated with or without AMF under drought stress has been shown as Figure 3 in lines 250-253. As follows:

Figure. 3 Effect of arbuscular mycorrhizal fungus (AMF) on the Plant height of C. migao seedlings.CK, non-AMF control; C.etunicatum, plants inoculated with C.etunicatum; F.mosseae, plants inoc-ulated with F.mosseae; Co-inoculation, plants inoculated with F.mosseae & C.etunicatum.

Yours sincerely,
Xiao-feng Liao
Forestry College, Research Center of Forest Ecology, Guizhou University
Huaxi District, Guiyang 550025, China
Tel: 86+13608500675
E-mail: karst0623@163.com

Reviewer 2 Report (New Reviewer)

General comments:

I reviewed the paper entitled “Arbuscular Mycorrhizal Fungi alter the growth, water status, and nutrient uptake of Cinnamomum migao and the soil nutrient stoichiometry under drought stress and recovery”.  The paper described the effect of individual and co-inoculation of  Funneliformis mosseae and Claroideoglomus etunicatum, in pot experiments, on growth, water status, and nutrient uptake of C. migao as well as the soil nutrient stoichiometry under well-watered (WW) and drought stress (DS) conditions. The idea of the manuscript is interesting; however, the methodologies are insufficient. I would have expected to see more professional experiments to confirm the role of AMF in alleviating the deleterious effect of drought on plants in microcosms. Also, I wish the authors highlight the main mechanisms induced by AMF to improve plant fitness under drought stress. Generally, the writing style and English language of the manuscript need major revision by a native speaker. Several confusing sentences must be clarified. I have a main concern about the number of samples, whereas the authors depended only on 6 replicates by only 6 plants which is a limited number of samples. I cannot accept the manuscript in this form for publication in the journal of Fungi, as needs major revision in terms of methodologies, presentation, and writing.  Therefore, I consider it a major revision.

Title:

I suggest “Arbuscular Mycorrhizal Fungi improve the growth, water status, and nutrient uptake of Cinnamomum migao and the soil nutrient stoichiometry under drought stress conditions”

Abstract:

Generally, the abstract is too much abbreviated and does not contain a conclusion and future work prospective

Specific Comments:

-          Lines: 20-21: Support your statement with numbers and data

-          Lines: 21-24: Unclear, reformate and avoid long sentences.

-          Lines: 21-24 Explain what is CK and TN as they are the first time to mention

-          Lines 24: Remove “And”

-          Line 24-26: Reformate to be more informative and clearer

Introduction

The introduction lacked the objectives of this research paper

Lines 52: remove “And”

Line 54: remove “Medicinal plants have seen increasing interest in recent times and in many parts of the world [19, 20]”

Line 56-57: Reformate “and it has been classified as near-threatened in the Red List of Biodiversity 56 in China: Volume of Higher Plants [23].

Line 57: remove “Clearly”

Line 65: Cite a reference after “when soil P is limited, roots and AMF selectively allocate more C and P to each other.”

Line 61: Cite a reference after “water supplying and nutrient cycling”

Line 72: Lauraceae is italicized

Materials and methods

Several parts are very confusing and need clarification. There are many grammatical mistakes that must be corrected. I have a main concern about the number of samples (only 6 plants which are very limited).  

Specific Comments:

-          The pot experiment was carried out in a greenhouse. I do not understand why the authors wrote a separates title as a “Study area”. Normally this is described for field experiments.

-          Also, it seems a controlled greenhouse, so why the authors mentioned: “the average

-annual rainfall is 1000–1400 mm, with 157 d of rainfall and 1000–1400 h of sunshine”!!!!!!!!!

-          Please ensure that latitude and longitude values are correct because when I checked them, I found nothing on Google Earth.

-          Lines 90-92: the soil physicochemical characteristics are a part of the results NOT materials and methods. Also, Indicate your reference for the soil analysis.

-          Line 94: Mycorrhizal fungal inocula

-          Line 95: were placed

-          Line 99: delete “which was “

-          Lines 114-115: individual and co-inoculation of ………

-          Line 118: a 20 g (dry wt) mycorrhizal??? This is not scientifically sound, please indicate the total number of spores.

-          Line 119: autoclaved inoculant???Explain why??

-          Lines 121-130: Confusing, reformate and avoid using long sentences.

-          Line 168: SOC???

Results

In general, the results section needs a major English revision. It is unclear and very confusing to the reader. It should be reformatted to be more interesting and attract the reader's attention. Several sentences are confusing, and I found it difficult to understand what the authors mean. The authors should focus on the important findings instead of repeating the data presented in the tables and figures. Please use the terminology “co-inoculated“ instead of “mixed”

-           Line 192, was significantly affected by the type of AMF and water treatment

-          Lines 192-195: unclear reformate

-          Line 205: remove “During the experimental period”

Discussion

Generally, the discussion is descriptive and needs careful rephrasing. The authors failed to provide a clear discussion for the obtained results. I do not recommend to divide the discussion into titles.

-          Line 425-426: “the mycorrhizal colonization rate of C. etunicatum plants was over-all highest than that of other treatments.“ Unclear and confusing

-          Line 426: Glomus is italicized

-          Line 430: Cite a reference “Thirdly, C. etunicatum is probably a native partner of C. migao and has been symbiotic for a long history

-          Line 431:  “Additionally, AMF morphology differed with family and genus of host plants, as well as with fungal identities” What does this sentence provide here?

-          434-436: Please support each reason with more references. Also, there are many reasons not only One reason.

-          Co-inoculation NOT mixed inoculation

-          439-443. This is not a clear explanation for the advantages of co-inoculation over single inoculation. Please review.

-           

Figures:

Figs.1-6

The resolution is very low and it is unclear

Tables

Table 1.

Write Duncan's letter as superscript

Author Response

Dear professor:

Re: Manuscript ID: jof-2187413 and Title: Arbuscular Mycorrhizal Fungi alter the growth, water status, and nutrient uptake of Cinnamomum migao and the soil nutrient stoichiometry under drought stress and recovery.

Hope this email finds you well.

We deeply appreciate your positive evaluation of our work. Thanks very much. And thank you for your precious comments and advice. Those comments are all valuable and very helpful for revising and improving our paper, as well as the important guiding significance to our researches. We have studied comments carefully and have made correction which we hope meet with approval. The main corrections in the paper and the responds to your comments are as flowing in a different color (blue).

Title:

I suggest “Arbuscular Mycorrhizal Fungi improve the growth, water status, and nutrient uptake of Cinnamomum migao and the soil nutrient stoichiometry under drought stress conditions”.

Response: We are extremely grateful to your title suggestion. Now the title has been changed to the new one “Arbuscular Mycorrhizal Fungi improve the growth, water status, and nutrient uptake of Cinnamomum migao and the soil nutrient stoichiometry under drought stress and recovery” according to your nice advice.

Abstract:

Generally, the abstract is too much abbreviated and does not contain a conclusion and future work prospective.

Response: We are extremely grateful to you for pointing out this problem. Now the abstract has been reformatted according to your professional advice in lines 13-33. As follows:

 “Abstract: Drought greatly influences the growth and ecological stoichiometry of plants in arid and semi-arid regions such as karst areas, where Cinnamomum migao (C. migao) is an endemic tree species that is used as a bioenergy resource. Arbuscular mycorrhizal fungi (AMF) play a key role in the nutrient uptake in the soil-plant continuum, increasing plant tolerance to drought. However, few studies have examined the contribution of AMF in improving the growth of C. migao seedlings and the soil nutrient stoichiometry under drought stress conditions. A pot experiment was conducted under natural light in a plastic greenhouse to investigate the effects of individual inoculation and co-inoculation of AMF [Funneliformis mosseae (F. mosseae) and Claroideoglomus etunicatum (C. etunicatum)] on the growth, water status, and nutrient uptake of C. migao as well as the soil nutrient stoichiometry under well-watered (WW) and drought stress (DS) conditions. The results showed that compared with non-AMF control (CK), AM symbiosis significantly stimulated plant growth and had higher dry mass. Mycorrhizal plants had better water status than corresponding CK plants. AMF colonization notably increased the total nitrogen and phosphorus content of C. migao seedlings compared with CK. Mycorrhizal plants had higher leaf and stem total carbon concentration than CK. The results indicated that AM symbiosis protect C. migao seedlings against drought stress through improving the growth, water status, and nutrient uptake. In general,the C. migao seedlings that formed with C. etunicatum showed most beneficial effect on the plant growth, water status, and nutrient uptake among all treatments. In the future, we should study more about the biology characteristics of each AMF in the field study to understand more ecological responses of AMF under drought stress, which can better provide meaningful guidance of afforestation projects in karst regions.”

  1. Lines: 20-21: Support your statement with numbers and data

Response: We are extremely grateful to your suggestion. Now the abstract has been reformatted according to your professional advice in lines 13-33, and some unclear statements have been removed.

  1. Lines: 21-24: Unclear, reformate and avoid long sentences.

Response: We are extremely grateful to your suggestion. Some unclear and long sentences have been removed and reformatted according to your professional advice.

  1. Lines: 21-24 Explain what is CK and TN as they are the first time to

mention.

Response: Thank you very much for your insightful comments. We have changed the “CK” to “non-AMF control (CK)” in abstract. And “TN” was replaced with “the total nitrogen content” too.

  1. Lines 24: Remove “And”

Response: Thank you very much for your comments. We have removed “And” in lines 24 according to your advice.

  1. Line 24-26: Reformate to be more informative and clearer.

Response: Thank you very much for your comments. Some unclear sentences have been removed and reformatted according to your advice.

Introduction

The introduction lacked the objectives of this research paper.

Response: Thank you very much for your precious comments and advice. The working objective of this research was added according to your advice in lines 80-81. As follows:

“The purposes of our study were to evaluate the effect of AMF on the growth, water status, and nutrient uptake of C. migao.”

  1. Lines 52: remove “And”

Response: Thank you very much for your comments. We have removed “And” in lines 58 according to your advice.

  1. Line 54: remove “Medicinal plants have seen increasing interest in recent times and in many parts of the world [19, 20]”

Response: Thank you very much for your comments. We have removed the sentence in line 59 according to your advice.

  1. Line 56-57: Reformate “and it has been classified as near-threatened in the Red List of Biodiversity 56 in China: Volume of Higher Plants [23].

Response: Thank you very much for your comments. We have removed the sentence in lines 58-60 and replaced as a new sentence “The wood of C. migao is used as a bioenergy resource, and the tree is also used as an ethnic medicinal plant [18]. However, many natural populations of C. migao have dis-appeared because of the allelopathy and autotoxicity [19, 20].”

  1. Line 57: remove “Clearly”

Response: Thank you very much for your advice. We have removed the “Clearly”.

  1. Line 65: Cite a reference after “when soil P is limited, roots and AMF selectively allocate more C and P to each other.”

Response: Thank you very much for your advice. In line 70 a new reference was cited after “when soil P is limited, roots and AMF selectively allocate more C and P to each other [32].”

  1. Kaur, S.& Suseela, V. Unraveling arbuscular mycorrhiza-induced changes in plant primary and secondary metabolome. Metabolites, 2021, 10(8). 335. DOI:10.3390/metabo10080335.
  2. Line 61: Cite a reference after “water supplying and nutrient cycling”

Response: Thank you very much for your advice. In lines 64-65 two new references were cited after “play a significant role in ecosystems through water supplying and nutrient cycling [26, 27]”

  1. Guo, Y., Gao, P., Li, F., & Duan, T. Y. Effects of AM fungi and grass endophytes on perennial ryegrass Bipolaris soro-kiniana leaf spot disease under limited soil nutrients. European Journal of Plant Pathology, 2019, 154(3), 659–671. https://doi.org/10.1007/s10658-019-01689-z.
  2. Li, F., Guo, Y., Christensen, M., Gao, P., Li, Y., & Duan, T. An arbuscular mycorrhizal fungus and Epichloë festucae var. lolii reduce Bipolaris sorokiniana disease incidence and improve perennial ryegrass growth. Mycorrhiza, 2018, 28(16), 159-169. https://doi.org/10.1007/ s00572-017-0813-9.
  3. Line 72: Lauraceae is italicized

Response: Thank you very much for your advice. We have italicized “Lauraceae” in line 77.

Materials and methods 

Several parts are very confusing and need clarification. There are many grammatical mistakes that must be corrected. I have a main concern about the number of samples (only 6 plants which are very limited).

Response: Thank you very much for your advice and suggestion. In our future experiments, we will consider as many samples as possible according to your advice.

  1. The pot experiment was carried out in a greenhouse. I do not understand why the authors wrote a separates title as a “Study area”. Normally this is described for field experiments. Also, it seems a controlled greenhouse, so why the authors mentioned: “the average annual rainfall is 1000–1400 mm, with 157 d of rainfall and 1000–1400 h of sunshine”!!!!!!!!!

Response: Thank you very much for your suggestion. We are so sorry some relevant description was missed because of our negligence. Actually, we conducted the experiment in a plastic greenhouse under natural light, which cannot control the temperature and humidity.

In lines 87-90, the sentences were rewritten as follows:

“We conducted the experiment from early April to early September 2021 in a plastic greenhouse under natural light with a day/night mean temperature of 24/16°C, mean humidity 57.3%, and average illumination: 1178.6 h·y−1. The plastic greenhouse was located at the Guizhou University in Guiyang City…”

  1. Please ensure that latitude and longitude values are correct because when I checked them, I found nothing on Google Earth.

Response: Thank you very much for your precious suggestion. We are so sorry some inaccurate descriptions confused you because of our negligence. In lines 90-91, the sentence was reformatted as follows: “Southwest China (26◦ 340 N, 106◦ 420 E; elevation: 242–1020 m).”

  1. Lines 90-92: the soil physicochemical characteristics are a part of the results NOT materials and methods. Also, Indicate your reference for the soil analysis.

Response: Thank you very much for your precious suggestion. We are so sorry some inaccurate descriptions confused you because of our negligence. In lines 98-100, the soil physicochemical characteristics we mentioned are the growth substrate instead of the soil specimen.

We reformatted the sentences in lines 95-100. As follows:

“Field soil collected from karst areas with severe rocky desertification (25◦64 N,107◦07 E; elevation: 790.28 m) was used as growth substrate for the experiment. The growth substrate was sieved (2 mm) and then sterilized for 4 h at 121°C and 0.14 MPa to eliminate all microorganisms. The growth substrate had the following chemical character-istics: pH 7.09 ± 0.56 (measured in water, 1:5 w/v); SOM (24.06 ± 1.40 g·kg−1); Total nitrogen content (3.30 ± 0.13 g·kg−1); Total phosphorus content (0.99 ± 0.05 g·kg−1); Total potassium content (16.02 ± 1.80 g·kg−1).”

  1. Line 94: Mycorrhizal fungal inocula

Response: Thank you very much for your advice. We have modified “Mycorrhizal fungus inocula” to “Mycorrhizal fungal inocula” in line 103.

  1. Line 99: delete “which was “

Response: Thank you very much for your advice. We have deleted “which was” in line 108.

  1. Lines 114-115: individual and co-inoculation of ………

Response: Thank you very much for your advice. We have rewritten the sentence in line 123 according to your advice. As follows: “four mycorrhizal treatments individual and co-inoculation of F. mosseae and C. etunicatum…”

  1. Line 118: a 20 g (dry wt) mycorrhizal??? This is not scientifically sound, please indicate the total number of spores.

Response: Thank you very much for your advice. We have rewritten the sentence in line 128 according to your advice. As follows: “Either 20 g (dry wt) mycorrhizal with nearly 1400 spores…”

  1. Line 119: autoclaved inoculant???Explain why??

Response: Thank you very much for your advice. We are so sorry some inaccurate descriptions confused you because of our negligence. Autoclaved inoculum for 4 h at 121°C and 0.14 MPa to eliminate all microorganisms.

In lines 124-125, the sentence was reformatted and modified as follows: “a sterilized inoculum (121◦C, 0.14 MPa for 4 h) of F. mosseae and C. etunicatum evenly was applied into the substrate as the non-AMF control (CK).”

  1. Lines 121-130: Confusing, reformate and avoid using long sentences.

Response: Thank you very much for your comments. We have rewritten the sentences in line 131-137 according to your advice. As follows:

“This was a dynamic experiment conducted in three stages: all of the plants were well-watered for the first 90 days prior to stress (PS) to ensure that the AM colonization rate reached a stable point, then they were subjected to drought stress (SS) for the next 30 days (seedlings began wilting), and then rewatered (REC) for the next 30 days (seedlings resumed growth). During the WW treatment, deionized water was supplied to keep the relative soil water content at 75% of filed capacity by weighing the pots every day. During the DS treatment, the pots were without water supply until seedlings began wilting.”

  1. Line 168: SOC???

Response: Thank you very much for your comments. We are so sorry some inaccurate descriptions confused you because of our negligence. We have modified “SOC” to “The total concentration of carbon (SOM)” in line 175.

Results

  1. Line 192, was significantly affected by the type of AMF and water treatment

Response: Thank you very much for your advice. We are so sorry some inaccurate descriptions confused you because of our negligence. We have modified “by AMF and DS” to “by the type of AMF and water treatment” in line 202 according to your advice.

  1. Lines 192-195: unclear reformate

Response: Thank you very much for your comments. We have removed the sentence in lines 203-204 and replaced as a new sentence “AMF colonization rate of C. etunicatum plants was significantly higher than that of F. mosseae and Co-inoculation plants regardless of the water regime.”

  1. Line 205: remove “During the experimental period”

Response: Thank you very much for your advice. We have removed the “During the experimental period” in line 215.

Discussion

  1. Line 425-426: “the mycorrhizal colonization rate of C. etunicatum plants was over-all highest than that of other treatments. “Unclear and confusing

Response: Thank you for the suggestion. We are very sorry for the mistakes in this manuscript. Now, we have modified the sentence to a new one “The mycorrhizal colonization rate of C. etunicatum plants was the highest in all treatments.” in lines 421-422.

  1. Line 426: Glomus is italicized

Response: Thank you very much for your advice. We have italicized “Glomus” in line 422.

  1. Line 430: Cite a reference “Thirdly, C. etunicatum is probably a native partner of C. migao and has been symbiotic for a long history

Response: Thank you very much for your advice. In lines 427-428 a new reference was cited after “Thirdly, C. etunicatum is probably a native partner of C. migao and has been symbiotic for a long history [56]”

  1. Köhl L, van der Heijden MGA. Arbuscular mycorrhizal fungal species differ in their effect on nutrient leaching. Soil Biology and Biochemistry, 2016, 94: 191-199. DOI10.1016/j.soilbio.2015.11.019.
  2. Line 431: “Additionally, AMF morphology differed with family and genus of host plants, as well as with fungal identities” What does this sentence provide here?

Response: Thank you very much for your advice. We are so sorry this sentence confused you because of our negligence. We have removed it now.

  1. Lines 434-436: Please support each reason with more references. Also, there are many reasons not only One reason.

Response: Thank you very much for your advice. Some relevant references were cited in lines 429-431. As follows:

“May be that drought enhances soil O2 [58], increases soil heterogeneity [59], and restricts nutrient availability [60] and mobility, providing more favorable conditions for the growth of aerobic microorganisms [61].”

  1. Tatsumi, C., Taniguchi, T., Du, S., Yamanaka, N., & Tateno, R. Soil nitrogen cycling is determined by the competition between mycorrhiza and ammonia‐oxidizing prokaryotes. Ecology. 2020, 101(3): e02963. 10.1002/ecy.2963.
  2. Murugesan, Chandrasekaran, Mak, Chanratana, Kiyoon, & Kim, et al. Impact of arbuscular mycorrhizal fungi on photosynthesis, water status, and gas exchange of plants under salt stress a meta-analysis. Frontiers in plant science. 2019, doi:10.3389/fpls.2019.00457.
  3. Marschner, P., & Baumann, K. Changes in bacterial community structure induced by mycorrhizal colonization in split-root maize. Plant Soil. 2003, 251(2), 279-289. Doi:10.1023/A:1023034825871.
  4. Aslam, M. M., Idris, A. L., Zhang, Q., et al. Rhizosphere microbiomes can regulate plant drought tolerance. Pedosphere. 2022, 32, 61-74. doi: 10.1016/s1002-0160(21)60061-9.
  5. Co-inoculation NOT mixed inoculation

Response: Thank you very much for your advice. We have replaced “mixed inoculation” with “Co-inoculation” in the manuscript according to your advice.

  1. Lines 439-443. This is not a clear explanation for the advantages of co-inoculation over single inoculation. Please review.

Response: Thank you very much for your advice. We have reviewed that and added a new explanation as follows:

“One reason may be that co-inoculation enhances interspecific competition and inhibits the increase of spore number in rhizosphere soil [63, 64].”

  1. Yang G., Guo LP., Guo XH., et al. Selectivity Infection of Arbuscular Mycorrhizal Fungi in Medicinal Plants. Chinese Journal of Information on TCM. 2012, 19(1):53-55. DOI: 10.3969/j.issn.1005-5304.2012.01.019 (in Chinese with English abstract).
  2. Wei ZH., Zhao SX., Li ZW., et al. Effects of Mixed Inoculation of Different Arbuscular Mycorrhizal Fungi on Saussurea Costus Rhizosphere Microorganisms and Soil Enzyme Activities. Chinese Wild Plant Resources. 2021, 40(8):6-11. doi10.3969/j.issn.1006-9690.2021.08. 002. (in Chinese with English abstract).

Figures:

  1. Figs.1-6

The resolution is very low and it is unclear

Response: We are extremely grateful to you for pointing out this problem. Now all figures were replaced with high resolution new figures according to your professional advice.

Tables

  1. Table 1.

 Write Duncan's letter as superscript

Response: Thank you for the suggestion. All Tukey 's letter have been written as superscript according to your professional advice.

Thank you very much for your attention and consideration!
Yours sincerely,
Xiao-feng Liao
Forestry College, Research Center of Forest Ecology, Guizhou University
Huaxi District, Guiyang 550025, China
Tel: 86+13608500675
E-mail: karst0623@163.com

Round 2

Reviewer 2 Report (New Reviewer)

Line 119: autoclaved inoculant

I believe you mean autoclaving the carrier NOT inoculant. 

Inoculant means the carrier plus microbes

Please check

Author Response

Dear professor:

Re: Manuscript ID: jof-2187413 and Title: Arbuscular Mycorrhizal Fungi improve the growth, water status, and nutrient uptake of Cinnamomum migao and the soil nutrient stoichiometry under drought stress and recovery.

Hope this email finds you well.

Many thanks for your insightful comments and suggestions of the referees. Those comments are valuable and very helpful. We have read through comments carefully and have made corrections. Appended to this letter is our point-by-point response to your comments. The comments are reproduced and our responses are given directly afterward in a different color (blue).

  1. Line 119: autoclaved inoculant

I believe you mean autoclaving the carrier NOT inoculant.

Inoculant means the carrier plus microbes

Please check.

Response: Thank you very much for your insightful suggestion. We have checked and changed the “autoclaved inoculant” to “sterilized inoculum”, and rewritten the sentence in lines 124-126 according to your advice as follows:

“…and a sterilized inoculum (121◦C, 0.14 MPa for 4 h to prevent spontaneous mycorrhization) of F. mosseae and C. etunicatum evenly was applied into the substrate as the non-AMF control (CK).”

Yours sincerely,
Xiao-feng Liao
Forestry College, Research Center of Forest Ecology, Guizhou University
Huaxi District, Guiyang 550025, China
Tel: 86+13608500675
E-mail: karst0623@163.com

This manuscript is a resubmission of an earlier submission. The following is a list of the peer review reports and author responses from that submission.

Round 1

Reviewer 1 Report

Scientific name in title should be italic.

In line 13, a coma is misused.

In the abstract, the introduction is too long, while the result and the method has not been described enough.

In line 94 and 96, the unit of the spore concentration (70/g, 120/g) was not sufficient.

The term “well-watered” has not been fully defined.

I believe the study was not appropriate to practice. The water use for the soil was deionized, while farmers usually use tap water. The soil needed to be well watered 90 days for the fungi to grow.

There should be a treatment where the fungi are not watered 90 days prior. If there has been irrigation, it cannot be called as drought stress.

How the rhizosphere soil was collected was not described?

 In figure 2, it is difficult to distinguish the columns, their colors should be changed.

Figure 3 should be divided into three different diagrams.

Author Response

Dear professor:

Re: Manuscript ID: jof-2121632 and Title: Arbuscular Mycorrhizal Fungi alter the growth, water status, and nutrient uptake of Cinnamomum migao and the soil nutrient stoichiometry under drought stress and recovery.

Merry Christmas!

Hope this email finds you healthy.

Many thanks for your insightful comments and suggestions of the referees. Those comments are valuable and very helpful. We have read through comments carefully and have made corrections. Appended to this letter is our point-by-point response to your comments. The comments are reproduced and our responses are given directly afterward in a different color (blue).

  1. Scientific name in title should be italic.

Response: Thank you for the title suggested. The precedent version of the title has been replaced, the new title is that “Arbuscular Mycorrhizal Fungi alter the growth, water status, and nutrient uptake of Cinnamomum migao and the soil nutrient stoichiometry under drought stress and recovery.”

  1. In line 13, a coma is misused.

Response: Thanks for your insightful suggestions. We have replaced the old word “affects” with a new one “influences” in line13.

  1. In the abstract, the introduction is too long, while the result and the method has not been described enough.

Response: Thanks very much for your insightful suggestions. We have rewritten the abstract according to your suggestion. As follows:

“Drought, greatly influences the growth and ecological stoichiometry of plants in arid and semi-arid regions such as karst areas, where Cinnamomum migao (C. migao) is an endemic tree species that is used as a bioenergy resource. Arbuscular mycorrhizal fungi (AMF) play a key role in the nutrient uptake in the soil-plant continuum, increasing plant tolerance to drought. A greenhouse pot experiment was conducted to investigate the effects of inoculation with Funneliformis mosseae (F. mosseae), Claroideoglomus etunicatum (C. etunicatum), and F. mosseae and C. etunicatum (Mixed) on the growth, water status, and nutrient uptake of C. migao as well as the soil nutrient stoichiometry under well-watered (WW) and drought stress (DS) conditions. AMF treatments increased the growth, water status, and P concentrations and the C and N concentrations in the seedlings. The water status of C. etunicatum plants was highest in all treatments. C. etunicatum plants exhibited greater leaf dry mass (195.95%), stem dry mass (175.41%), and root dry mass (91.94%) compared with the CK seedlings and greater root TN values of 30.14, 43.50, and 32.14% compared with the CK. And the results of a correlation analysis indicated that there were strongly coupled relationships between the plants and soil ecological stoichiometries.”

  1. In line 94 and 96, the unit of the spore concentration (70/g, 120/g) was not sufficient.

Response: Thank you very much four your insightful comments. We have changed the unit to the new one “70 spores/g, 120 spores/g” in line 94 and 96.

  1. The term “well-watered” has not been fully defined.

Response: Thank you for your suggestion very much. We are so sorry some relevant description was missed because of our negligence. It has been added in line 115-116. “The experiment was conducted in three stages: all of the plants were well-watered (WW) (control with 75% of relative soil water content by weighing the substrate before and after drying at 105℃ for 24 h)”.

  1. I believe the study was not appropriate to practice. The water use for the soil was deionized, while farmers usually use tap water. The soil needed to be well watered 90 days for the fungi to grow.

Response: Thank you for your profound insight and precious comments. The purpose of our present study is that exploring which AMF can more effectively associate with Cinnamomum migao seedlings and improve their drought resistance. And then, we will conduct a field study to verify the AMF we have selected in our present study so as to have more practical significance.

  1. There should be a treatment where the fungi are not watered 90 days prior. If there has been irrigation, it cannot be called as drought stress.

Response: Thank you very much for your precious suggestion. In our future experiments, we will consider the treatment you suggested. Our present study is a dynamic one, all treatments are well watered 90 days prior to ensure that the AM fully inoculate with Cinnamomum migao seedlings.

  1. How the rhizosphere soil was collected was not described?

Response: Thank you for the suggestion. We are so sorry some relevant description was missed because of our negligence. It has been added in line 127-129. “After removed the top 2 cm of topsoil, the firmly attached soil (1-2 mm distance from the root) were collected and regarded as rhizosphere soil, respectively.”

  1. In figure 2, it is difficult to distinguish the columns, their colors should be changed.

Response: Thank you for the suggestion. We have changed the colors in figure 2. Simultaneously, we have changed the colors in figure S1 and figure S2 which are difficult to distinguish the columns too. Thanks very much.

  1. Figure 3 should be divided into three different diagrams.

Response: Thank you for your suggestion. We have divided Figure 3 into three different diagrams (Figure3, Figure 4 and Figure 5).

Yours sincerely,
Xiao-feng Liao
Forestry College, Research Center of Forest Ecology, Guizhou University
Huaxi District, Guiyang 550025, China
Tel: 86+13608500675
E-mail: karst0623@163.com

Reviewer 2 Report

Dear,

The paper “Arbuscular Mycorrhizal Fungi alter the growth, water status, and nutrient uptake of Cinnamomum migao and the soil nutrient stoichiometry under drought stress and recovery”, although not very innovative, provides some relevant data.

Several adjustments are required:

1) Scientific names of plants and fungi must be written in italics;

2) Do not start a sentence with the abbreviated genus of organisms;

3) The first time the scientific name appears in the text, the name of the species classifier must be presented; next times, the genus is abbreviated and there is no need to mention the classifier;

4) The introduction should be reformulated, bringing more relevant data on mycorrhizae and the aspects studied. It is very important to add the working hypothesis;

5) Inform about the AMF inoculum: substrate used for multiplication, host, inoculum storage data, infectivity potential and fungal cultivation data (temperature, air relative humidity and global solar radiation);

6) The results are well presented, but the discussion fails to explain the reason for the different behavior of each AMF tested: it is important to understand the biology of each inoculated fungus, especially in drought conditions. This is the main thing to be addressed in this item. In this sense, this section must be reformulated, considering the approach of the work hypothesis (refuted or corroborated).

7) The conclusions resemble final considerations... review.

Sincerely, 

Author Response

Dear professor:

Re: Manuscript ID: jof-2121632 and Title: Arbuscular Mycorrhizal Fungi alter the growth, water status, and nutrient uptake of Cinnamomum migao and the soil nutrient stoichiometry under drought stress and recovery.

Merry Christmas!

Hope this email finds you healthy.

We deeply appreciate your positive evaluation of our work. Thanks very much. And thank you for your precious comments and advice. Those comments are all valuable and very helpful for revising and improving our paper, as well as the important guiding significance to our researches. We have studied comments carefully and have made correction which we hope meet with approval. The main corrections in the paper and the responds to your comments are as flowing in a different color (blue).

  1. Scientific names of plants and fungi must be written in italics.

Response: We are extremely grateful to you for pointing out this problem. All scientific names of plants and fungi in the manuscript have been written in italics according to your nice advice.

  1. Do not start a sentence with the abbreviated genus of organisms.

Response: Thank you for the suggestion. We are very sorry for the mistakes in this manuscript. Now, we have checked all sentence and revised them in line50, 52.

  1. The first time the scientific name appears in the text, the name of the species classifier must be presented; next times, the genus is abbreviated and there is no need to mention the classifier.

Response: Thank you for underlining this deficiency. We have checked all the scientific names in the manuscript, and changed them according to your suggestion. Thanks very much.

  1. The introduction should be reformulated, bringing more relevant data on mycorrhizae and the aspects studied. It is very important to add the working hypothesis.

Response: Thank you very much for your precious comments and advice. We have reformulated the introduction and added more relevant references in line 63-68, simultaneously, the working hypothesis was added according to your advice. As follows: Arbuscular mycorrhizal fungi (AMF) are beneficial microorganisms in the soil associated with the roots of 70-90% of terrestrial plant species. AMF play a significant role in ecosystems through water supplying and nutrient cycling, processes that strongly influence biogeochemical cycles of C, N, and P. AMF acquire nutrients from the soil then transfers to the host plants in exchange for photosynthetically fixed carbon (C). Simultaneously, 5-10% of photosynthetically of host plants is allocated to AMF partner, when soil P is limited, roots and AMF selectively allocate more C and P to each other. Under drought stress, AM fungi can absorb and transport water and N to host plants more efficient through hypha. AMF enhance their host plants’ tolerance to biotic and abiotic stresses, especially drought. Studies have demonstrated that AMF can increase the plant yield and help plants to resist the biotic and abiotic stresses that occur in agriculture and forestry, making the symbiotic interactions of plants with mycorrhizal fungi agriculturally and ecologically important. Several studies have revealed that AMF are abundant in karst areas and that they can form a symbiosis with a range of Lauraceae species, including C. migao. However, little is known about the effects of AMF on C. migao growth, water status, or C, N, and P ecological stoichiometry under drought stress and recovery in karst areas. We hypothesized that: (i) AMF inoculation can positively affect the growth and drought resistance of C. migao under drought stress conditions. (ii) AMF inoculation can positively contribute to the nutrient uptake especially P of C. migao.

  1. Inform about the AMF inoculum: substrate used for multiplication, host, inoculum storage data, infectivity potential and fungal cultivation data (temperature, air relative humidity and global solar radiation)

Response: Thank you very much for your precious comments and advice. The information about the AMF inoculum has been added in line 94-99 according to your suggestion. “5 g of each inocula with white clover white clover (Trifolium repens) seeds was placed in a pot (22.5 cm diameter × 22.5 cm height) which was filled with 2 kg sterilized sand (126◦C, 0.14 MPa for 4 h) respectively. And then the pot was placed in a temperature incubator (25 ◦C, 75% relative humidity, 3000 lx (16 h light/8 h dark)) for 4 months. After removing the above-ground parts of the plants and 2 cm of topsoil, soil and underground parts of plants are collected as inocula. ”

  1. The results are well presented, but the discussion fails to explain the reason for the different behavior of each AMF tested: it is important to understand the biology of each inoculated fungus, especially in drought conditions. This is the main thing to be addressed in this item. In this sense, this section must be reformulated, considering the approach of the work hypothesis (refuted or corroborated)

Response: Thank you very much for your precious comments and advice. Firstly, the purpose of our present study is that exploring which AMF can more effectively associate with Cinnamomum migao seedlings and improve their drought resistance. Second, we've ploughed through all the relevant literatures according to your advice, regrettably few studies have researched the biology characteristics of each AMF. We are so sorry that We're afraid we can't explain this phenomenon at the moment. In the future, we should try our best to do more about the biology characteristics of each AMF and understand ecological responses and effects of AMF under drought stress. The discussion has been rewritten in line 402-415 according to your suggestion and our guesses which based on the literatures. As follows:

“One reason is that Glomus species are typical of semi-arid ecosystems, even in karst areas, and are able to adapt and grow under drought stress. Second, the main ecological function of C. etunicatum to promote plant primary production and P absorption, especially in the karst regions, where P is the main limiting element. Thirdly, C. etunicatum is probably a native partner of C. migao and has been symbiotic for a long history. Additionally, AMF morphology differed with family and genus of host plants, as well as with fungal identities. C. etunicatum may have longer hypha than other fungi for adapting to drought. For each AMF, the mycorrhizal colonization rate was significantly increased under drought stress. One reason may be that drought enhances soil O2, increases soil heterogeneity, and restricts nutrient availability and mobility, providing more favorable conditions for the growth of aerobic microorganisms. Moreover, when exposing to stress, plant roots can detect the environ-mental signals and release more secondary metabolome to attract microorganisms”.

  1. The conclusions resemble final considerations... review

Response: Thank you very much for your precious comments and advice. The conclusion has been rewritten in line 537-548 according to your suggestion. As follows: “In summary, the present study has demonstrated that AMF can improve the growth, water status, and nutrient uptake of C. migao seedlings under drought stress and recovery. It is clear that AMF promotes phosphorus uptake for host plants so that more C and N can be allocated to the leaves and roots and AMF always play a key role in the nutrient uptake of host plants by regulating soil nutrient contents under drought stress and recovery. It is obvious that C. etunicatum had most beneficial effect on the plant growth, water status, and nutrient uptake among all treatments. Many of the current studies have found this phenomenon and that few related studies are conducted with the underlying mechanisms still unknown. In the future, we should study more about the biology characteristics of each AMF and understand ecological responses and effects of AMF under drought stress. We should also progress more field study, which can better provide detailed and meaningful guidance of afforestation projects in Karst regions.

Thank you very much for your attention and consideration!
Yours sincerely,
Xiao-feng Liao
Forestry College, Research Center of Forest Ecology, Guizhou University
Huaxi District, Guiyang 550025, China
Tel: 86+13608500675
E-mail: karst0623@163.com
